# Radiomics-Based Prediction of Treatment Response to TRuC-T Cell Therapy in Patients with Mesothelioma: A Pilot Study

**DOI:** 10.3390/cancers17030463

**Published:** 2025-01-29

**Authors:** Hubert Beaumont, Antoine Iannessi, Alexandre Thinnes, Sebastien Jacques, Alfonso Quintás-Cardama

**Affiliations:** 1Median Technologies, 06560 Valbonne, France; antoine.iannessi@mediantechnologies.com (A.I.); alexandre.thinnes@mediantechnologies.com (A.T.); sebastien.jacques@mediantechnologies.com (S.J.); 2TCR2 Therapeutics, Cambridge, MA 02142, USA; dragbona@gmail.com

**Keywords:** clinical trial, mesothelioma cancer, pilot study, radiomics, response criteria

## Abstract

T cell receptor fusion constructs (TRuC-Ts) represent a promising next-generation T cell therapy for solid tumors. To enhance their development, improved patient selection is essential. A pilot study was conducted to evaluate the feasibility and performance of a predictive model for treatment responses in mesothelioma patients, leveraging radiomics and machine learning. Radiomics and delta-radiomics (Δradiomics) features from CT scans were analyzed for reproducibility and informativeness, identifying key features for training a random forest classifier. The model achieved an accuracy of 0.75–0.9 in predicting pleural tumor responses, supporting the design of future studies involving 250–400 tumors. This study demonstrated the reproducibility and effectiveness of radiomics/Δradiomics in relation to tumor localization, emphasizing the need for multiple tumor models to create an integrated patient model. These findings provide a foundation for combining tumor-specific models into a unified approach, improving patient selection for TRuC-T therapy in mesothelioma patients.

## 1. Introduction

T cell receptor fusion constructs (TRuCs) represent a next generation engineered T cell platform endowed with the ability to target cancer antigens by leveraging the power of the entire T cell receptor (TCR) in an HLA-independent manner [1]. Despite the remarkable success of chimeric antigen receptors (CARs) in several hematological malignancies, multiple studies have failed to replicate similar results among patients with solid tumors, which has hindered their adoption in that setting [2,3]. While novel platforms such as TRuCs hold great promise for the treatment of solid tumors, improving the benefit/risk ratio of these therapies will require an improved upstream selection of patients to improve response rates and minimize treatment-related toxicities.

Several criteria can be considered for evaluating therapeutic responses with imaging techniques. The Response Evaluation Criteria for Solid Tumors (RECISTs) [4] are the most commonly used criteria in oncology clinical trials, although some limitations have been pointed out, particularly their ability to assess certain types of tumors and to evaluate several unique tumor response subgroups outside of a response classification by RECIST, such as a mixed response and an oligometastatic state [5,6]. An alternative method to RECIST, which is based on the assessment of tumor diameter, is that of evaluating tumor volume. Several groups have described the better sensitivity and reproducibility of tumor volume measurements, and the Quantitative Imaging Biomarker Alliance (QIBA) [7] is currently in the final phase of validation for changes in lung tumors volume assessed through computed tomography (CT). On the other hand, positron emission tomography (PET) imaging offers a different insight into therapeutic responses with the quantification of a standard uptake value (SUV) to assess tumor metabolic changes, although it is not exempt from toxicity such as cumulative radiation exposure [8]. So far, none of these different paradigms of responses have clearly outperformed the others. Therefore, analyzing the three of them would provide a wider, more precise, and perhaps complementary, tumor response assessment.

Over the past decade, radiomics, a high-throughput imaging technique, has shown to be effective in detecting and predicting treatment responses [9] as well as in gauging the subtle changes in a tumor microenvironment not adequately addressed by conventional imaging analyses [10].

Radiomics is an innovative approach to medical imaging that identifies patterns that are invisible to the human eye but can be correlated with biological features and clinical outcomes. Radiomics uses high-dimensional data from radiological images to measure basic morphological features (e.g., tumor volume), and the distribution of voxel intensity (e.g., derived from a histogram) to their joint distributions like the Grey Level Co-occurrence Matrix (GLCM), but also a diversity of other indexes derived from the grey-level run length matrix, the neighborhood grey-level difference matrix, or shape-related features. Frequential analysis (e.g., wavelet, Gabor) is being increasingly used nowadays [11]. Furthermore, while radiomics analyzes a static situation, delta-radiomics (Δradiomics) provide an analysis of feature variation over time, i.e., extracting radiomic features from the same region of interest in the same patient at different acquisition time points [12].

The quantitative features extracted by radiomics can be used to predict tumor and patient outcomes at a given time point, whereas Δradiomics assesses feature variation over different acquisition periods, such as before and after therapy [13]. In patients with mesothelioma, PET radiomic features have been shown to be associated with progression-free survival and overall survival rates [14], whereas such an association has not been identified by a standard CT imaging evaluation. To date, no Δradiomics study has been reported on patients with mesothelioma with any imaging method.

The objective of this pilot study was to evaluate whether radiomics or Δradiomics have the potential to predict the response of patients with mesothelioma in the context of a clinical trial where patients were treated with gavocabtagene autoleucel (gavo-cel), a TRuC T cell therapy targeting mesothelin. We evaluated the reproducibility of radiomics and Δradiomics and their correlation with the therapeutic responses of target lesions, as well as the performance of a machine learning algorithm. These different outcomes were used to provide a power analysis and to suggest a classification scheme.

## 2. Material and Methods

### 2.1. Study Data

This retrospective study included 23 patients with advanced mesothelin-expressing mesothelioma treated in a phase 1/2, open label, single arm clinical trial (NCT03907852). In this clinical trial, the patients were assessed by CT and PET imaging in a double read Blinded Independent Central Review (BICR) setting. Imaging evaluations were derived from the estimation of the overall response rate (ORR), which included a complete response (CR) and a partial response (PR), according to RECIST v1.1 evaluation criteria. In addition to the regular one-dimensional measurement stipulated in RECIST, volume measurements were performed in all selected target tumors. The same imaging modality and image-acquisition protocols (including the use of an intravenous contrast) were consistently used at all time points for each patient to ensure uniformity in the comparison of lesions.

In addition, the PET data were analyzed according to PERCIST criteria [15] considering the SUV of the hottest tumor at each time point.

### 2.2. Definition of the Response

At the target tumor level, three different definitions of responses were considered: (1) stable (+25%, −30% change) or a decrease (−30% decrease) in target tumor diameters (RECIST thresholds) with CT imaging; (2) at least a decrease in tumor volume as defined by QIBA [16] with CT imaging; and (3) at least 30% of the hottest mean tumor SUV with PET (PERCIST threshold). The subjective response of non-target lesions was not considered. For each response definition, the ratio between the number of responding tumors and the total number of tumors was computed as the response rate.

### 2.3. Radiomic Feature Evaluations

The original CT-based tumor volume evaluations were extracted from the iSee platform (Median Technologies, Valbonne, France), then converted to Nifti format. Radiomic features were calculated using LifeX (https://www.lifexsoft.org/ (accessed on 15 December 2024)) freeware [17].

Delta-radiomics were calculated as the net change between the baseline evaluations and the first time point after baseline using the following formula [18]:*NetChange* = *Feature_Time point 1_* − *Feature_Baseline_*

For the reproducibility study, the relative change in response evaluations was calculated using the following formula:*RelativeChange* = (*Feature_Timepoint1_* − *Feature_Baseline_*)/*Feature_Baseline_*

### 2.4. Study Plan

Data from the original clinical trial were stratified according to patients’ disease and type of tumor (Figure 1).

#### 2.4.1. Reproducibility of Tumor Measurements

Because the original trial featured a double reading setting (i.e., two independent blinded reviewers) for each patient at each time point, a subset of target tumor measurement could be paired. For each radiomic and Δradiomic, the reproducibility was assessed by measuring the paired difference of the measurements.

Previous studies reported that reproducibility can differ according to tumor localization [19,20]. In our study, we hypothesized that reproducibility is homogeneous within each tumor localization, notably between the responding and non-responding tumors. We documented the variability per tumor localization and tested the equivalence of the inter-localization reproducibility.

For each tumor localization, we compared the variability of each radiomic and Δradiomic feature with those reported in the literature [21,22].

#### 2.4.2. Univariate Analysis

We performed multiple comparisons of the mean radiomics values at baseline, week 4 (W4), W8, and W12 by tumor type. For each radiomic and Δradiomic, we performed univariate analyses by testing the equivalence of the mean values for the responding and non-responding tumors (according to diameter, volume, and mean SUV Ground Truth (GT)) at W4, W8, and W12.

#### 2.4.3. Features Selection

We considered an original set of 3416 radiomics, from which feature selection was required [23]. The selection strategy consisted of four steps: (1) a selection of the most reproducible radiomics and Δradiomics determined by a reproducibility (univariate) analysis; (2) a removal of the redundant radiomics and Δradiomics determined by a correlation coefficient threshold; (3) a removal of radiomics and Δradiomics after a process of recursive feature elimination; and (4) a final selection of the number of radiomic and Δradiomic features recommended to avoid data overfitting.

#### 2.4.4. Model Design for Predicting Response to Treatment

A per-tumor model was trained in processing radiomics or Δradiomics independently for each main tumor localization: pleura, lymph nodes, peritoneum, and soft tissues.

#### 2.4.5. Statistics

Statistics were processed using R CRAN software (V. 4.3.3) [24]. Statistical significance levels were two-sided, with *p*-values < 0.05 and a 95% confidence interval (CI). Because of the small sample size, non-parametric statistics were preferred.

Sunburst display was performed using the “SunburstR” package (V. 2.1.8).

Reproducibility:

We evaluated reproducibility of each radiomic and Δradiomic feature using concordance correlation coefficients (CCCs) as defined by Lin [25]. For several threshold values of concordance, we reported the number of radiomic features higher than those thresholds. The *p*-value associated with significant non-inferiority values over the thresholds was tested with a bootstrapping CCC (“boot” and “boot.pval” packages) and adjusted for multiple tests using the “Fuzzysim” package. Then, we compared inter-tissue reproducibility by applying an F-Test for the equality of two variances [26] between the two most prevalent types of tumors and by comparing their CCC CIs.

Univariate analysis:

We tested the non-equivalence of the values across tumor localizations for each radiomics/Δradiomics using multiple Kruskal–Wallis tests with a false discovery rate (FDR) correction (Fuzzysim package) [27].

Typical values for each type of tumor were provided with a CI using the “interpretCI” package.

If non-equivalence of the radiomics values across tumor localizations was found, the tumor localizations were studied separately.

We tested whether radiomics/Δradiomics were associated with the responding/non-responding tumors using the Kruskal–Wallis test with an FDR correction (Fuzzysim package).

Cross-radiomics correlations were measured using the Spearman’s rank correlation coefficient (a non-parametric counterpart to Pearson’s correlation). Redundant features were deleted when correlation coefficients were >0.9.

Features selection:

Features reduction was processed by the following:Selecting the reproducible radiomics/Δradiomics based on CCC values (as previously mentioned).Selecting non-redundant radiomics/Δradiomics using a Spearman’s rank correlation coefficient of 0.9. We relied on the clustering [28] package “heatmaply” and the “fmradio” package for display and to perform data reduction after clustering.Running a recursive feature elimination (RFE) based on the random forest algorithm.Avoiding overfitting issues by establishing an acceptable maximal number of predictors in line with the previous pilot studies [20]. The *p*-to-n ratio was held at 10.

Model design:

Binary responses were classified using the “caret” package in a cross-validation setting with a random forest algorithm [29].

Proportional and corresponding CIs were calculated using the Clopper–Pearson exact CI model from the “PropCI” package, while multiple comparisons of the proportions (e.g., tumor localization) were computed using the Marascuilo test [30].

## 3. Results

### 3.1. Imaging Data

The distribution of patients by modality (CT and PET) and center (*n* = 5) at baseline is depicted in Figure 2.

The number of patients with CT images taken was 23, 22, 18, and 13 at baseline, W4, W8, and W12, respectively, while there were 17, 17, 5, and 9 patients at baseline and the subsequent time points, respectively, with PET images taken.

Figure 3 summarizes in sunburn the distribution of the CT and PET acquisition parameters used by the five imaging centers.

### 3.2. Anatomical Distribution of Tumors

The anatomical distribution of the target tumors at baseline in the 23 patients is depicted in Figure 4.

The distribution of target tumors by patient is summarized in Table 1. Overall, 39.1% (9/23) of patients had a single tumor localization.

### 3.3. Criteria of Response—Corresponding Response Rate

For the main tumor localizations, the response rate at each time point is presented by the tumor diameter (responding or stable disease), tumor volume, and mean SUV in Table 2, Table 3 and Table 4, respectively.

### 3.4. Reproducibility

Double assessments were performed on 21 patients, for whom 67 pairs of tumors were measured at baseline and at follow-ups. Tumors were distributed in the pleura (*n* = 35), lymph nodes (*n* = 14), soft tissues (*n* = 12), and peritoneum (*n* = 5). Figure 5 shows the number of radiomics and Δradiomics with the highest CCC values by tumor localization.

For the pleura and lymph node tumors, which were the most prevalent tumors, 41.5% of radiomic features had a significantly different variability (*p* < 0.05, F-Test) and the volume CCCs were 0.36 (95% CI: 0.31; 0.41) and 0.64 (95% CI: 0.30; 0.84), respectively.

For the pleural and lymph node tumors, 39.1% of Δradiomic features had a significantly different variability (*p* < 0.05, F-Test) and the volume CCCs were 0.57 (95% CI:0.43; 0.68) and 0.18 (95% CI: −0.03; 0.38), respectively.

### 3.5. Univariate Analysis

#### 3.5.1. Inter-Tumor Differences in Features Values

In total, 16.0% (95% CI: 15.0%; 17.0%) (*n* = 536) and 10.0% (95% CI: 9.0%; 11.0%) (*n* = 338) of the radiomic and △radiomic features, respectively, have different values across tumor type.

After correction for multiple testing (FDR, q = 0.05), these proportions were 2.7% (95% CI: 2.1%; 3.2%) (*n* = 90) and 0% (*n* = 0), respectively.

At baseline, the tumor volumes were 62.3 (95% CI: 18.8; 105.8) cm^3^, 17.8 (95% CI: 5.9; 29.8) cm^3^, 34.5 (95% CI: 12.0; 57.1) cm^3^, and 20.2 (95% CI: 6.3; 34.0) cm^3^ in the pleura, lymph node, peritoneum, and soft tissue tumors, respectively. We found no significant differences between tumor volumes (*p* = 0.6).

#### 3.5.2. Association Between Radiomics and Responses

For each radiomic/Δradiomic and the main tumor localizations, we tested whether the responder/non-responder populations were significantly different. We tested three paradigms of responses: diameter-based, volume-based, and PERCIST, and the results are presented in Table 5, Table 6 and Table 7, respectively. After FDR correction (q = 0.05), no radiomics/Δradiomics were associated with responses in the diameter, volume, or PERCIST.

Number of radiomic features that had a significant difference of means between the responder/non-responder (Kruskal–Wallis test). Some subcategories were not evaluated (NA) because of the limited response rate.

### 3.6. Feature Selection and Model Design

To avoid data overfitting, the “10 samples per predictor” rule of thumb is commonly used. Therefore, our model should not rely on more than approximately four, two, two, and two predictors for classifying pleura (*n*= 47), lymph node (*n* = 15), peritoneum *n* = 12), and soft tissue (*n* = 9) tumors, respectively. We selected a maximum of three predictors for classifying the pleural tumors and two for the other tumors.

#### 3.6.1. Preselection

Considering the reproducible radiomic/Δradiomic features (Figure 5a,b), we removed those with an inter-correlation value >0.9 as shown in the sample cluster map in Figure 6.

The number of radiomics/Δradiomics candidates are depicted in Figure 7a,b.

#### 3.6.2. Feature Selection/Models Performances

The responses of the target pleural tumors were predicted based on three radiomics (three wavelets) measured at baseline. The accuracy was 0.9 (95% CI: 0.6; 0.95), the area under the curve (AUC) = 0.88, at W8, and the accuracy was 0.75 (95% CI: 0.34; 0.96), with an AUC = 0.74, at W12.

Peritoneum tumors were predicted based on two radiomics with an accuracy of 0.67 (95% CI: 0.1; 0.99) and an AUC = 0.61.

The other subclasses (of tumor type/visit) could not be evaluated because of the cross-correlation setting that split the dataset, leaving the test set with missing responder or non-responder data.

The responses of the pleural tumors were predicted based on three Δradiomics (one wavelet and two from density percentiles). The accuracy was 0.71 (95% CI: 0.3; 0.9), with an AUC = 0.74, at W8, and 0.8 (95% CI: 0.3; 0.99), with an AUC = 0.68, at W12. The other subclasses (of tumor type/ visit) could not be tested due to limitations in cross-validation.

The predictions of the SUV responses of the pleura tumors were based on two radiomics (2 wavelets); the accuracy was 0.71 (95% CI: 0.3; 09), with an AUC = 0.68 at W4 and 0.75 (95% CI: 0.2; 0.9), with an AUC = 0.7 at W12. Based on two Δradiomics (two from density percentiles), the prediction had an accuracy of 0.75 (95% CI: 0.2; 0.9) and an AUC = 0.75 at W12.

### 3.7. Power Analysis

Considering the two most reproducible and informative radiomics and Δradiomics drawn from the feature selection, we estimated the sample size needed to reach significancy in setting shrinkage value at 0.9.

Considering the volume-derived responses and three radiomics predictors at W8, with an AUC = 0.8 and a 40% response rate, the minimum sample size would be 369 patients with 148 responses. At W12, with an AUC = 0.7 and a 27% response rate, the minimum sample size would be 303 patients with 82 responses. When considering two Δradiomics predictors with an AUC = 0.7 and a 30% response rate, the minimum sample size would be 323 patients with 97 responses at W8. At W12, with an AUC = 0.8 and a 30% response rate, the minimum sample size would be 323 patients with 97 responses.

Regarding the SUV-derived responses, when considering two radiomics predictors, at W4, with an AUC = 0.8 and a 20% response rate, the minimum sample size would be 246 patients with 50 responses. At W12, with an AUC = 0.7 and a 40% response rate, the minimum sample size would be 369 patients with 148 responses. Using two Δradiomics predictors (CCC = 0.6), with an AUC = 0.75 and a 36% response rate, the minimum sample size would be 355 patients with 128 responses at W12.

## 4. Discussion

### 4.1. Staging, Distribution, and Response Biomarkers

This pilot study documented the anatomical distribution of tumors in patients with mesothelioma and showed the imbalance between the responding/non-responding tumors across location and imaging biomarkers, including longest diameter, volume, and mean SUV.

There was a strong imbalance between the responding/non-responding tumors, which significantly limited the value of a diameter-derived imaging assessment paradigm in favor of volume-derived response criteria using CT as the imaging modality.

Importantly, more than half of the target tumors evaluated in patients with mesothelioma were pleural-based, with the lymph nodes, peritoneum, and soft tissues being the second most frequent sites of disease involvement, in line with the reported patterns of metastasis in patients with malignant pleural mesothelioma described by Collins et al. [31]. Collins et al. also concluded that metastatic spread did not appear to have prognostic implications for overall survival. In our study, brain lesions were excluded, and bone lesions were not selected as a “target” in compliance with RECISTv1.1 guideline recommendations. Additionally, a subset of patients had pleural tumors, but they were not selected as a target for a radiomics analysis, partially due to the RECIST selection criteria requiring a “measurable” supra-centimetric lesion.

Radiomics Robustness

In line with previous studies [32,33], we found that the average radiomics/Δradiomics values varied depending on tumor localization. Therefore, from the standpoint of the average value and reproducibility, different sets of radiomics should be considered for tumor applications according to their localization. Consequently, the models aimed at predicting patient responses should integrate a tumor-specific classification and, for the final system to be practical and widely applicable, it must transition from a tumor-centered approach to a patient-centered one. This would require the implementation of hierarchical or multi-level modeling to ensure the system can address the broader complexities of patient care.

For a predictive model, a higher number of reproducible radiomics/Δradiomics allows more flexibility for feature selection, and thus for training. However, in this study, a small proportion of radiomics/Δradiomics were deemed reproducible and the number of reproducible tumor radiomics/Δradiomics were significantly different between the different anatomic locations, with soft tissue and lymph node tumors featuring the highest and lowest number of reproducible radiomics, respectively. Previous studies have shown that the variability of segmentation is a significant factor impacting radiomics reproducibility [34]. Figure 8 illustrates how the complexity of tumor segmentation can impact radiomics variability.

In contrast to Zhao et al. [21], we found that the volume of pleural tumors had a low CCC value, which might be explained by several factors. First, while the data evaluated by Zhao et al. were lung data, the pleural tumors data of this pilot study were probably more complex to segment. Second, the dataset average volume was largely different between the two studies (around 22.0 cm^3^ and 62.3 cm^3^ for the lung data and this study, respectively). Even if a lower relative error can be expected for the larger segmentations, these are more likely to be adjacent to other anatomical structures and, therefore, are prone to segmentation errors. In addition, it is worth noting that pleural mesothelioma involvement causes a particular pattern of growth and spread, with circumferential pleural spread being more prevalent than the typical nodular growth observed in most solid tumors. Consequently, the identification of target lesions with a nodular morphology can be challenging in patients with mesothelioma and adds a layer of variability between two independent blinded observers (Figure 8). For this reason, modified RECIST criteria adapted to the evaluation of pleural mesothelioma have been developed [35].

### 4.2. Radiomics Correlations

While no radiomics or Δradiomics were associated with tumor responses in the univariate analyses, a multivariate classification allowed us to predict the responses of pleural tumors.

Since a comprehensive evaluation of our predictive system was applied to document the distribution of responses for each tumor location (*N* = 4) and time point (*N* = 3), the response criteria would ideally feature an acceptable sample size and balance between the responder/non-responder for each subclass. We considered three different criteria for tumor responses, two for CT (tumor diameter and volume) and one for PET (mean SUV). In agreement with Cai et al. [36], we observed that tumor diameter had a different sensitivity/specificity than the volume measurements in CT. The responses derived from the CT diameter measurements were deemed suboptimal for a prediction assessment because of the strong imbalance between the responding/non-responding tumors. Therefore, we focused on the response in volume with CT and on the mean SUV with PET.

In our univariate analysis, some of the 3416 radiomics were correlated to the responses before an FDR correction. After an FDR correction, none of the radiomics or Δradiomics correlated to any response criteria, as previously reported by Chalkidou et al. [37], thereby supporting the use of multivariate models and machine learning.

Our feature selection scheme aimed to select a maximum of two to three radiomics/Δradiomics per tumor localization to avoid data overfitting in a cross-validation setting. Significant classification performances were obtained only for the pleural tumors with an accuracy ranging from 0.9 to 0.75, which could power future studies involving a data sample size of 250 to 400 target tumors.

### 4.3. Radiomics and Study Limitations

Several study limitations are worth discussing. First, due to the retrospective nature of our study, we had to adapt the measurements from the original trial (NCT03907852), which prevented us from relying on more specific response criteria [35]. Therefore, we applied the standard RECIST 1.1 criteria, and in particular, the volume-derived response criteria, to all mesothelioma tumors regardless of their anatomical location, even though the volume was originally qualified for advanced lung disease. Although QIBA’s profile [16] clearly highlights the limitations of extending the use of volumetry to other diseases, we adopted these criteria because of the high level of qualification.

Second, we grouped all the tumors into four main localizations. The tumors were initially labeled by each independent reader, then grouping was subjectively performed on a radiologic basis by a single third-party radiologist. Aware that this process was prone to inter-group and intra-group errors [38], we attempted to balance the limited sample size and our hypothesis that each tumor localization had a different average value and reproducibility.

Third, avoiding data overfitting was a challenge due to the small sample size of some subclasses. No well-established statistics are currently available for sizing the maximum number of predictors. However, since the univariate analysis showed no correlation, the minimum number of predictors (2–3) was selected to minimize the risk of data overfitting.

Fourth, we were uncertain about the complexity and potential non-linear relationships between the features and target variables. Considering the small sample size, we aimed to minimize the risk of overfitting and the impact of noise on the data. Therefore, we chose Random Forest as a robust and flexible classifier. Although we considered testing other classification methods, comparing algorithm performance was beyond the scope of our study. Fifth, one aspect that was not addressed in this study is how to translate the models designed at the target tumor level into a patient predictive system. We analyzed the correlation and prediction of target tumor responses using the radiomics/Δradiomics values at baseline since these measurements can only be obtained for measurable tumors (target lesions). However, the real value of a predictive system lies in its ability to assist with patient management, which brings up the issue of response criteria. The response evaluation by the RECIST criteria combines the assessment of target, non-target, and new tumors. However, our analysis was performed only on target lesions and, therefore, such analysis would require the use of validated radiomics-derived, target-lesion-only-based response criteria, which is yet to be developed. Such response criteria will need to address the cases where the responding patients have a dissociated response within a specific tumor site and, even more challenging, the cases where dissociated responses occur at different tumor sites.

Lastly, a proper radiomics study is expected to explain the outcome from the imaging and clinical point of view [39]. We did not address this point because it was considered out of the scope of a pilot study [40] and it could have led to premature conclusions.

### 4.4. Perspectives

The upcoming pivotal study is anticipated to validate the predictive performance of our pilot study. However, the radiomics-only approach presented here is just a foundational step that could be significantly enhanced by incorporating additional imaging biomarkers, such as nutritional biomarkers (e.g., L3—Skeletal Muscle Index) [41], as well as non-imaging biomarkers (e.g., Krebs von den Lungen-6 (KL-6)) [42]. Furthermore, advancements in AI models and the integration of complementary technologies could further elevate the system’s capabilities [43].

Given the ability of radiomics to predict the therapeutic response of target lesions, we could also start considering evaluating the potential of radiomics for clinical trial monitoring, thereby improving the current response criteria used for mesothelioma.

## 5. Conclusions

This study supports the use of radiomics/Δradiomics and machine learning for response prediction in patients with malignant mesothelioma. Future studies will have to be powered by assessing reproducibility as well as addressing the different response paradigms and their limitations. As the evidence shows that technologies can be efficient, additional resources and imaging data should be implemented in the evaluation of patients with mesothelioma. We believe that radiomics/Δradiomics and machine learning provide the means to address response prediction in patients with mesothelioma, which thus far has not been adequately addressed. In addition to the predictive model described herein, we propose that the analysis of responses per tumor location could also aid in better understanding drug efficacy in the future.

## Figures and Tables

**Figure 1 cancers-17-00463-f001:**
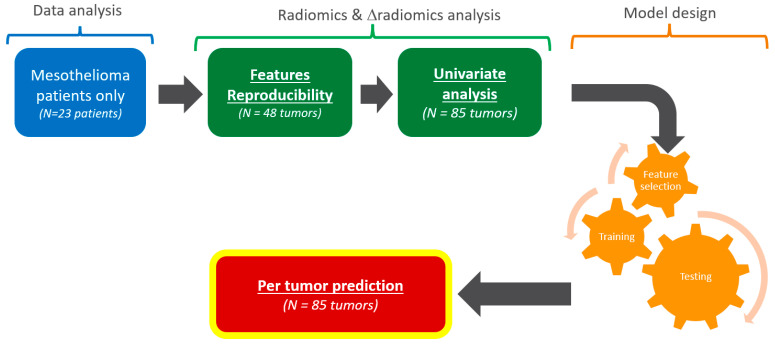
Analysis workflow. From the original clinical trial, only mesothelioma patients were considered. This study consisted of a data analysis (blue), radiomics and Δradiomics analysis (green), and model design (orange). The outcome was a pilot evaluation of the predictive performances of target tumors responses.

**Figure 2 cancers-17-00463-f002:**
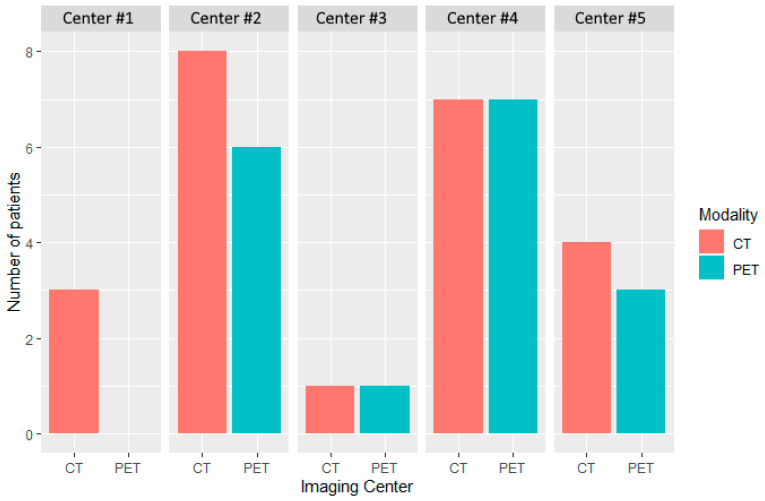
Distribution of patients by modality and center at baseline. Five imaging centers participated in the evaluation of patients treated in this study. CT and PET imaging were performed in 23 and 17 patients, respectively. A single imaging center performed only CT (Center #1).

**Figure 3 cancers-17-00463-f003:**
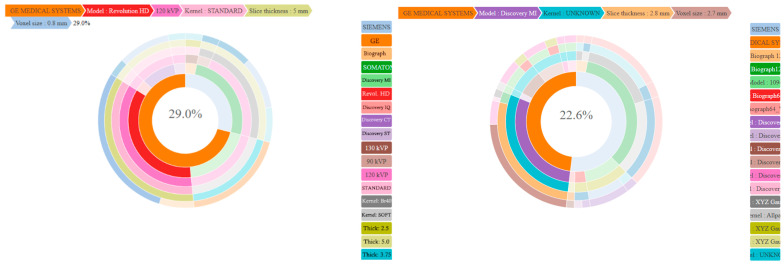
Distribution of acquisition parameters at baseline. (Left) CT acquisition parameters. From inner to outer circle: manufacturer (Siemens or GE), models, Kvp (90, 120, 130), reconstruction kernel (Standard, Br40, Soft), slice thickness (2.5; 5), and voxel size (0.6; 1.0). (Right) PET acquisition parameters. From inner to outer circle: manufacturers (Siemens (Siemens Healthineers, Forchheim, Germany), GE (GE Healthcare, Milwaukee, WI, US)), models, reconstruction kernel (AllPass, XYZ Gauss (2.0, 3.5, 5.0)), slice thickness (2.0; 5.5), and voxel size (1.5; 6.9). Representativeness was deemed significant for generalization.

**Figure 4 cancers-17-00463-f004:**
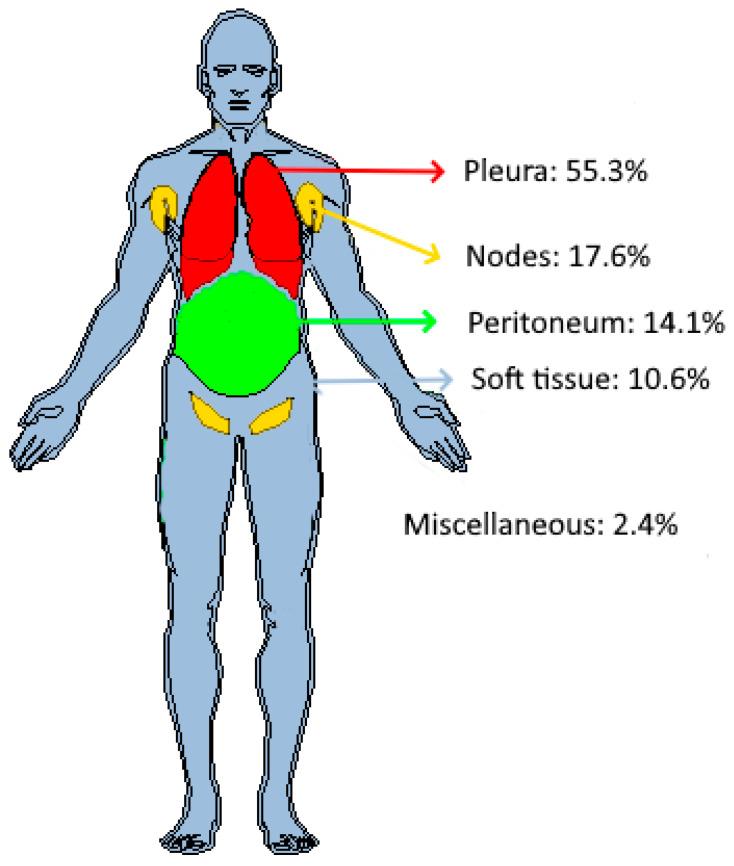
Distribution of target tumors in patients’ anatomy. Out of 85 tumors, 55.3% (*n* = 47) were found in the pleura, 17.6% (*n* = 15) in the lymph nodes, 14.1% (*n* = 12) in the peritoneum, and 10.6% (*n* = 9) in soft tissues. Two additional tumors, one adrenal tumor and one liver tumor, were classified as “miscellaneous”.

**Figure 5 cancers-17-00463-f005:**
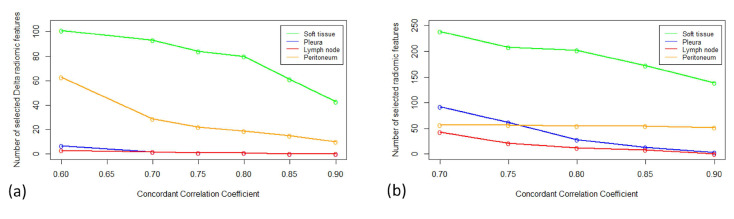
Radiomic/Δradiomic features according to different CCCs threshold values. The number of radiomics (**a**) and Δradiomics (**b**) were calculated for different threshold values of CCC. Radiomics reproducibility depended on tumor localization, with soft tissues (range: 238; 139) and lymph nodes (range: 43; 0) being the most and least reproducible, respectively. The reproducibility of Δradiomics depended on tumor localization, with soft tissues (range: 101; 53) and lymph nodes (range: 3; 0) being the most and least reproducible, respectively.

**Figure 6 cancers-17-00463-f006:**
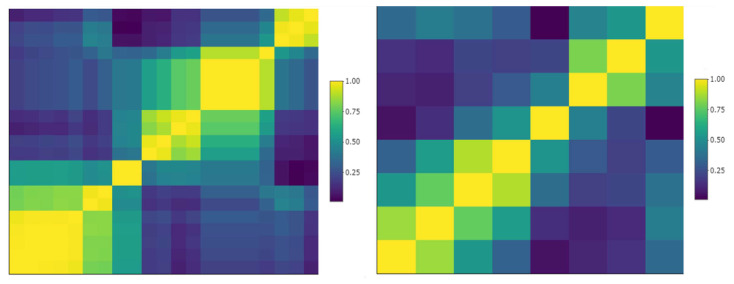
Sample cluster map of reproducible pleural tumors radiomics. Left: correlation matrix of 21 radiomics that were deemed reproducible (CCC > 0.8); some of them were highly inter-correlated (yellow clusters). Right: after removing highly inter-correlated radiomics (correlation > 0.9), 8 reproducible and non-redundant pleura radiomics were preselected.

**Figure 7 cancers-17-00463-f007:**
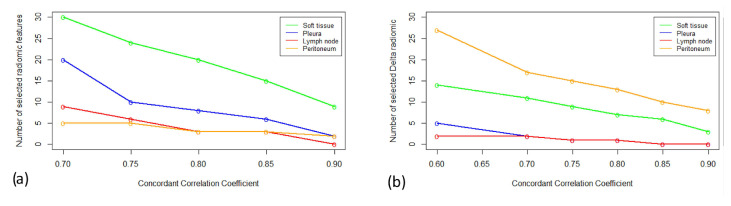
Number of radiomic/Δradiomic candidates. We considered a different threshold of CCC values ranging from 0.7 to 0.9 for radiomics (**a**) and 0.6 to 0.9 for Δradiomics (**b**). For radiomics and Δradiomics, peritoneum and lymph node tumors were the most and least reproducible tumors, respectively.

**Figure 8 cancers-17-00463-f008:**
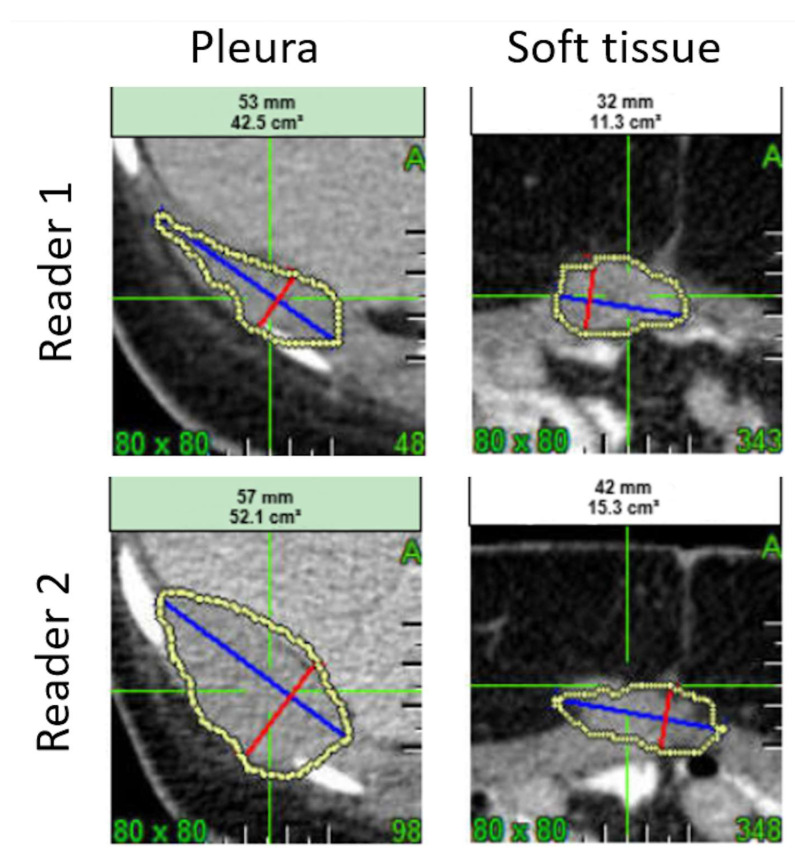
Sample variability of segmentation and radiomics values. One pleura and one soft tissue tumor were segmented by Reader 1 and Reader 2. The volume, the joint entropy, and the sum of variance (computed from GLCM) were derived from the segmentations. The inter-reader variability of the segmentations leads to a variability in volume of 20% and 30%, in joint entropy of 16% and 7%, and in sum of variance of 41% and 56%, respectively, the pleura and the soft tissue tumors.

**Table 1 cancers-17-00463-t001:** Distribution of target tumor localization by patient at baseline.

Tumor Localization	Disease in Patients	Numb. of Patient with a Unique Disease
Pleura	13	5
Lymph nodes	6	0
Peritoneum	8	3
Soft tissues	4	1
Miscellaneous	2	0

Rows report the main tumor localizations. Central column: number of patients for whom the corresponding tumor localization was reported. Right column: number of patients for whom a single tumor localization was reported.

**Table 2 cancers-17-00463-t002:** Response rate derived from tumor diameter. Response rate is presented by percentages and proportions. With a response rate of 100%, six subcategories (red) were unusable for analysis and model training.

.	Week 4	Week 8	Week 12
Pleura	83% (39/47)	95% (36/38)	97% (28/29)
Lymph nodes	80% (12/15)	91% (10/11)	100% (10/10)
Peritoneum	100% (12/12)	100% (8/8)	100% (2/2)
Soft tissues	100% (9/9)	100% (7/7)	71% (5/7)
All	87% (72/83)	95% (61/64)	94% (45/48)

**Table 3 cancers-17-00463-t003:** Response rate derived from tumor volume. Response rate is presented as percentages and proportions. Because of its limited sample size, one subcategory was unusable for analysis and model training (red).

.	Week 4	Week 8	Week 12
Pleura	30% (14/47)	32% (12/38)	28% (8/29)
Lymph nodes	50% (7/15)	55% (6/11)	33% (3/10)
Peritoneum	33% (4/12)	25% (2/8)	50% (½)
Soft tissues	33% (3/9)	28% (2/7)	15% (1/7)
All	34% (28/83)	34% (22/64)	27% (13/48)

**Table 4 cancers-17-00463-t004:** Response rate of Mean SUV tumors responses. Response rate is presented as percentages and proportions. Because of their limited sample size or inadequate response rate, some subcategories were unusable for analysis and model training (red).

.	Week 4	Week 8	Week 12
Pleura	20.5% (8/39)	58% (7/12)	36% (9/25)
Lymph nodes	64% (7/11)	100% (1/1)	14% (1/7)
Peritoneum	40% (4/10)	34% (2/6)	0% (0/1)
Soft tissues	0% (0/7)	0% (0/1)	0% (0/7)
All	67% (19/67)	50% (10/20)	25% (10/40)

**Table 5 cancers-17-00463-t005:** Radiomics associated with diameter-based response of target tumor. Number of radiomic features that had a significant difference of means between the responder/non-responder (Kruskal–Wallis test). Some subcategories were not evaluated (NA) because of the limited response rate.

Organ	Radiomics			ΔRadiomics	
	W4	W8	W12	W8	W12
Pleura	15 (*N* = 47)	7 (*N* = 38)	0 (*N* = 29)	81 (*N* = 38)	0 (*N* = 29)
Lymph nodes	76 (*N* = 15)	0 (*N* = 11)	NA	0 (*N* = 11)	NA
Peritoneum	NA	NA	NA	NA	NA
Soft tissues	NA	NA	1 (*N* = 7)	NA	0 (*N* = 7)

**Table 6 cancers-17-00463-t006:** Radiomic features associated with volume-based response of target tumors. Number of radiomic features that had a significant difference of means between the responder/non-responder (Kruskal–Wallis test). Some subcategories were not evaluated (NA) because of the limited response rate.

Organ	Radiomics			ΔRadiomics	
	W4	W8	W12	W8	W12
Pleura	185 (*N* = 47)	584 (*N* = 38)	290 (*N* = 29)	505 (*N* = 38)	221 (*N* = 29)
Lymph nodes	99 (*N* = 15)	93 (*N* = 11)	53 (*N* = 10)	404 (*N* = 11)	164 (*N* = 10)
Peritoneum	122 (*N* = 12)	55 (*N* = 8)	NA	0 (*N* = 8)	0 (*N* = 2)
Soft tissues	129 (*N* = 9)	2 (*N* = 7)	0, (*N* = 7)	2 (*N* = 7)	0 (*N* = 7)

**Table 7 cancers-17-00463-t007:** Radiomic features associated with PERCIST response of target tumors.

Organ	Radiomics		ΔRadiomics	
	W4	W12	W4	W12
Pleura	532 (*N* = 39)	209 (*N* = 25)	73 (*N* = 39)	71 (*N* = 25)
Lymph nodes	282 (*N* = 11)	53 (*N* = 7)	48 (*N* = 11)	** NA **
Peritoneum	89 (*N* = 10)	NA	278 (*N* = 10)	NA
Soft tissues	NA	NA	NA	NA

## Data Availability

The data that support the findings of this study are the property of TCR2 therapeutics. Restrictions apply to the availability of these data, which were used under license for this study.

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
