# Peer review of "Radiomics-Based Prediction of Treatment Response to TRuC-T Cell Therapy in Patients with Mesothelioma: A Pilot Study"

_cancers, 2025, doi:10.3390/cancers17030463_

Round 1
Reviewer 1 Report
Comments and Suggestions for Authors
I have the following questions about this paper:
1. The author needs to clarify the standards and procedures used to examine the three tumor response assessment paradigms. Since there is no obvious superiority among the paradigms, how will the author ensure that the analysis is thorough and able to spot possible complementarities?
2. How does the exclusion of non-mesothelioma patients and the exclusive focus on target tumor responses affect the predictive model's generalizability and applicability to a larger population, and what further steps are planned to validate the model across various clinical scenarios?
3. The decision to restrict the number of predictors for every tumor categorization should be supported by the author: Given the different sample sizes (N=47, N=15, N=12, and N=9), what standards or supporting data underlie the particular upper bounds of 4, 2, 2, and 2 predictors, and how do these limitations affect the model's performance?
4. The author may provide more details about the reasoning behind and application of tumor-specific classification.
5. The author may cover the possible impact of imaging center variability on the study's findings.
6. The effects of segmentation variability on repeatability across anatomical locations should be discussed by the author. What measures were taken to reduce segmentation variability during the investigation?
7. The author may elaborate on how the segmentation issues particular to particular tumor types were resolved: What steps were taken to guarantee precise and reliable radiomic analysis in these situations, and how were segmentation mistakes in bigger segmentations and the unique circumferential growth pattern of pleural mesothelioma taken into consideration in the study?
8. The author may discuss the effect of acquisition parameter variability on study generalizability.
9. How does concentrating on volume with CT and mean SUV with PET address the imbalance between responding and non-responding tumors, and what evidence supports the superiority of these metrics over diameter measurements for predictive accuracy?
10. The author may discuss the drawbacks of doing a target-lesion-only analysis and suggest a justification.
Author Response
Dear Sir,
Thank you for your thoughtful and insightful comments, notably regarding the variability due to acquisition parameters.
We have carefully addressed each of your comments in detail below, we emphasize the constraint of our retrospective study for which we depended on the initial NCT03907852 trial regarding the annotations. We hope that our responses will meet your expectations.
Best regards,
Comment 1: The author needs to clarify the standards and procedures used to examine the three tumor response assessment paradigms. Since there is no obvious superiority among the paradigms, how will the author ensure that the analysis is thorough and able to spot possible complementarities?
Answer 1: Our study was a retrospective investigation based on the original NCT03907852 trial, which adhered to RECIST 1.1 criteria with the addition of volume contouring. The NCT03907852 trial also complied with PERCIST criteria. All evaluations underwent rigorous quality control and monitoring, as it was a double-read trial.
We did not assume the superiority of any particular quantitative imaging biomarker (QIB), such as diameter, volume, or SUV, nor did we have any means to determine the absolute “truth” in this context. Investigating the complementarity of these paradigms would indeed be a fascinating study, but it was beyond the scope of this work. Instead, the focus of this pilot study was intentionally narrow: to explore whether certain radiomics features have the potential to predict changes in one of the three QIBs, without attempting to combine them.
Comment 2: How does the exclusion of non-mesothelioma patients and the exclusive focus on target tumor responses affect the predictive model's generalizability and applicability to a larger population, and what further steps are planned to validate the model across various clinical scenarios?
Answer 2: The original NCT03907852 study was specifically designed to evaluate mesothelioma patients. Consequently, the results of our retrospective study cannot be generalized beyond this specific indication.
Regarding the insightful question about our focus on target tumors, additional studies will indeed be necessary. In our current study, we assessed whether baseline radiomics features could predict changes in diameter, volume, or SUV at later time points. A complementary future study would involve predicting changes not only in target tumors but also incorporating non-target and new lesions, thereby generalizing the analysis to more comprehensive response criteria.
The objective of our study, as stated in the manuscript, was: “to evaluate whether radiomics or Δradiomics have the potential to predict the response of patients with mesothelioma in the context of a clinical trial where patients were treated with gavocabtagene autoleucel (gavo-cel), a TRuC T cell therapy targeting mesothelin.” The next step will be to confirm the predictive performance of the model at the tumor level in the pivotal study. Since the pivotal study will involve the same imaging centers, and therefore similar acquisition parameters, the subsequent step will be to validate the model using data from different imaging centers with varying acquisition parameters.
Finally, as detailed in Answer 4 and Answer 10, a hierarchical model with be evaluated to transition to tumor-specific into patient-specific model.
Comment 3: The decision to restrict the number of predictors for every tumor categorization should be supported by the author: Given the different sample sizes (N=47, N=15, N=12, and N=9), what standards or supporting data underline the particular upper bounds of 4, 2, 2, and 2 predictors, and how do these limitations affect the model's performance?
Answer 3: The rationale for using a variable number of predictors based on different sample sizes was primarily to prevent overfitting. In the “Statistics” section, under the “Feature Selection” subsection, we outlined our methodology and cited a relevant reference: “Avoiding overfitting issues by establishing an acceptable maximal number of predictors, in line with previous pilot studies [20]. The p-to-n ratio was held to 10.”
Additionally, in the “Results” section, under the “Feature Selection and Model Design” subsection, we emphasized: “To avoid data overfitting, the commonly used ‘10 samples per predictor’ rule of thumb was applied. Consequently, our model should rely on no more than approximately 4, 2, 2, and 2 predictors for classifying pleura (N=47), lymph nodes (N=15), peritoneum (N=12), and soft tissue (N=9) tumors, respectively. We selected a maximum of 3 predictors for pleural tumors and 2 predictors for the other tumor types.”
Comment 4: The author may provide more details about the reasoning behind and application of tumor-specific classification.
Answer 4: We fully agree that this is a crucial point. Our work should be seen as an initial step toward demonstrating the feasibility of such predictions. As we noted in the core manuscript, the next steps will involve developing a hierarchical or multi-level modeling approach and designing response criteria that account for new lesions and non-target lesions. Extensive answer to this point is provided in detail below in Answer 10, additional rational is provided in the manuscript (See Answer 10).
Comment 5: The author may cover the possible impact of imaging center variability on the study's findings.
Answer 5: The variability introduced by imaging centers is arguably one of the most challenging limitations in the field of radiomics. This issue is part of the broader challenge posed by the variability in acquisition parameters, which we address in detail below in Answer 8
Comment 6: The effects of segmentation variability on repeatability across anatomical locations should be discussed by the author. What measures were taken to reduce segmentation variability during the investigation?
Answer 6: In the section of our manuscript titled “Reproducibility of Tumor Measurements”, we outlined our strategy to adapt to segmentation variability rather than attempting to improve it. This approach assumes (and confirms) that different tumor locations exhibit varying levels of reliability in segmentation. This rationale led us to first develop a tumor-specific prediction system, which will be later expanded into a hierarchical, patient-specific prediction system. We provide further details on this point in our response to Comment 7
Comment 7: The author may elaborate on how the segmentation issues particular to particular tumor types were resolved: What steps were taken to guarantee precise and reliable radiomic analysis in these situations, and how were segmentation mistakes in bigger segmentations and the unique circumferential growth pattern of pleural mesothelioma taken into consideration in the study?
Answer 7: Due to the retrospective nature of our study, we were reliant on the settings and measurements established in the original clinical trial (NCT03907852). In this trial, RECIST 1.1 criteria was used, with volume contouring added as an exploratory measure. A rigorous quality control process and reader monitoring system were implemented during the original double-read trial. As a result, we were not permitted to modify any measurements or contouring and had to rely on the evaluations provided.
In the section of our manuscript titled “Reproducibility of Tumor Measurements”, we indirectly assessed the reliability of the double reads and the radiomics measurements derived from them. For prediction purposes, we selected only the most reliable types of radiomics features, ensuring that this selection process accounted for all sources of variability, including readers perceptions, image selection, and measurements (diameters and volume). This reliability-driven selection approach has been documented extensively in multiple publications (doi.org/10.3390/sym15101834), which now reflect a growing consensus.
Comment 8: The author may discuss the effect of acquisition parameter variability on study generalizability.
Answer 8: Thank you for bringing up this very interesting topic. In previous publications (doi: 10.1007/s00330-020-07641-8 and doi: 10.1007/s00330-021-08154-8), we evaluated various strategies for harmonizing CT data across imaging centers. Our conclusions were that while harmonization is essential, it carries significant risks. Since those publications, we can assume that several innovative techniques have emerged; however, these likely require extensive validation, especially for addressing subtle signals, such as those found in radiomics. However, we strongly believe that relative intra-scan harmonization is one of the promising solutions (doi.org/10.1038/s41598-019-57325-7).
That said, beyond harmonization techniques, we see two potential approaches:
- Include a highly diverse dataset to claim generalizable results.
- Restrict the range of acquisition parameters to limit inter-scan variability, ensuring reproducible radiomics responses.
Unfortunately, the second approach raises another question: What is the optimal range of acquisition parameters to reliably capture radiomics signals? At present, we do not have an answer to this question. Therefore, the only solution we have found is to describe our dataset as thoroughly and transparently as possible while advocating for generalizability to datasets like ours.
Comment 9: How does concentrating on volume with CT and mean SUV with PET address the imbalance between responding and non-responding tumors, and what evidence supports the superiority of these metrics over diameter measurements for predictive accuracy?
Answer 9: Thank you for giving us the opportunity to clarify this key methodological question. We did not claim any superiority of volume or SUV over diameter, nor do we have any means to determine the absolute “truth” in this context. Instead, our approach highlights that, regardless of the reference used (diameter, volume, or SUV), an imbalance in the data between responders and non-responders poses a significant challenge when analyzing changes of any kind. Given that the imbalance between responders and non-responders is most pronounced in the “diameter” dataset, we chose not to consider it in our analysis.
Comment 10: The author may discuss the drawbacks of doing a target-lesion-only analysis and suggest a justification.
Answer 10: We fully agree that this is a key point. Our work must be view as a first step to get evidence of the feasibility of such prediction, as we indicated in the core manuscript, the next steps will be to enable a hierarchical or multi-level modeling and to train it against a response criterion that would consider new lesions and non-target lesions.
In the “Conclusion” of the abstract, we announced:” Tumor-specific reproducibility and average values indicated that bridging tumor model to effective patient model would require combining several target tumors models. “. In the manuscript, in the “Radiomics robustness” section we emphasize the need to transition from a tumor-center system to a patient-centered system in adding: “and for the final system to be practical and widely applicable, it must transition from a tumor-centered approach to a patient-centered one. This would require the implementation of hierarchical or multi-level modeling to ensure the system can address the broader complexities of patient care.”. Also, in the “Limitation” section, we mentioned: “Fifth, one aspect that was not addressed in this study is how to translate models designed at target tumor level into a patient predictive system. We analyzed the correlation and prediction of target tumor response using radiomics/Δradiomics values at baseline since these measurements can only be obtained for measurable tumors (target lesions). However, the real value of a predictive system lies in its ability to assist patient management, which brings up the issue of response criteria.”

Reviewer 2 Report
Comments and Suggestions for Authors
Methodology:
I really appreciate that the authors added more details on the preprocessing and feature extraction steps. Even though this is phoneme and there are no specific details about normalization and harmonization methods, there are clear actions that the authors can do to deal with potential variability in radiomics features due to differences in imaging centers and acquisition parameters.
The authors highlight the limits in the response criteria applied, for example, QIBA criteria, for tumors that are mesothelioma. These unique aspects of mesothelioma would be ripe for discussion and require potentially different, more tailored approaches to response assessment, such as the revised modified RECIST criteria for malignant pleural mesothelioma.
Results:
Add ROC curves and scatter plots for better visualization of results. We, however, feel that the assessment of the agreement between the radiomics-based predictions and the ground truth response data could be done better by the authors by adding more graphical representations like the Bland-Altman plots.
Present key caveats for the nature of this field, because of a complete lack of monitoring for any properties of all the delta-radiomics studies makes conclusion drawing, even over speculated sources, rather hard to perform in this context; authors should henceforth look into additions for higher order quantitative analyses as a means of comprehensively envisioning feature reproducibility.
Discussion:
Besides, the authors have also provided additional information about possible clinical significance of the radiomics-based predictive model-that is a real quite helpful thing. Clinical considerations for implementation are mentioned by the authors but they do not cover this in detail. discuss potential benefit of artificial intelligence. cite doi: 10.1007/s00405-024-08511-5.
Insightful discussion of challenges in utilizing the tumor-specific models for a patient-level predictive system. More specific recommendations-for example, hierarchical or multi-level modeling approaches that might surmount some of the challenges faced by future studies.
Future Directions:
The authors went further to propose some relevant next steps, including the validation of the predictive model in an independent cohort and the use of radiomics/delta-radiomics in monitoring response to their treatment over time. Furthermore, the authors explored the possibility of integrating radiomics features with other clinical or molecular variables to enhance predictive performance.
Authors could have emphasized the need for developing and validating radiomics-derived, target-lesion-only-based response criteria beyond the currently accepted criteria for mesothelioma, which may more aptly address a few of the complexities of the disease process.
Author Response
Dear Sir,
Thank you for your thoughtful and insightful comments, which have significantly contributed to enhancing the clarity and quality of our manuscript, particularly in the newly added “Perspective” section.
We have carefully addressed each of your comments in detail below and hope that our responses meet your expectations.
Best regards,
Top of Form
Methodology:
Comment 1: I really appreciate that the authors added more details on the preprocessing and feature extraction steps. Even though this is phoneme and there are no specific details about normalization and harmonization methods, there are clear actions that the authors can do to deal with potential variability in radiomics features due to differences in imaging centers and acquisition parameters.
Answer 1: Thank you for bringing up this very interesting topic. In previous publications (doi: 10.1007/s00330-020-07641-8 and doi: 10.1007/s00330-021-08154-8), we evaluated various strategies for harmonizing CT data across imaging centers. Our conclusions were that while harmonization is essential, it carries significant risks. Since those publications, we can assume that several innovative techniques have emerged; however, these likely require extensive validation, especially for addressing subtle signals, such as those found in radiomics. However, we strongly believe that relative intra-scan harmonization is one of the promising solutions (doi.org/10.1038/s41598-019-57325-7).
That said, beyond harmonization techniques, we see two potential approaches:
- Include a highly diverse dataset to claim generalizable results.
- Restrict the range of acquisition parameters to limit inter-scan variability, ensuring reproducible radiomics responses.
Unfortunately, the second approach raises another question: What is the optimal range of acquisition parameters to reliably capture radiomics signals? At present, we do not have an answer to this question. Therefore, the only solution we have found is to describe our dataset as thoroughly and transparently as possible while advocating for generalizability to datasets similar to ours.
Comment 2: The authors highlight the limits in the response criteria applied, for example, QIBA criteria, for tumors that are mesothelioma. These unique aspects of mesothelioma would be ripe for discussion and require potentially different, more tailored approaches to response assessment, such as the revised modified RECIST criteria for malignant pleural mesothelioma.
Answer 2: Thank you for these comments. As noted, due to the retrospective nature of our study, we were dependent on the settings of the original clinical trial (NCT03907852). The original trial utilized RECIST 1.1, with volume contouring added for exploratory purposes. We have expanded on the first limitation by highlighting our constraint of relying on RECIST 1.1 and volume-based criteria rather than more tailored criteria for mesothelioma. This revision has been inserted (L531–536): “First, due to the retrospective nature of our study, we had to adapt the measurements from the original trial (NCT03907852), which prevented us from relying on more specific response criteria [35]. Therefore, we applied the standard RECIST 1.1 criteria, and in particular, the volume-derived response criteria, to all mesothelioma tumors regardless of their anatomical location, even though the volume was originally qualifyed for advanced lung disease”
Results:
Comment 3: Add ROC curves and scatter plots for better visualization of results. We, however, feel that the assessment of the agreement between the radiomics-based predictions and the ground truth response data could be done better by the authors by adding more graphical representations like the Bland-Altman plots.
Answer 3: Thank you for these valuable suggestions for improvement, which we fully agree with. In the early versions of the manuscript, we included a large number of illustrations—such as ROC curves, confusion matrices, and Bland-Altman plots—to demonstrate variability in both the response and certain radiomics values. However, this abundance of figures made the manuscript difficult to navigate and overly complex.
To enhance clarity, we decided to focus on more concise metrics, such as AUC, accuracy, and ICC. Even after streamlining, our current submission includes 8 figures and 7 tables, which is still a substantial number of illustrations, even by the standards of many journal editors
Comment 4: Present key caveats for the nature of this field, because of a complete lack of monitoring for any properties of all the delta-radiomics studies makes conclusion drawing, even over speculated sources, rather hard to perform in this context; authors should henceforth look into additions for higher order quantitative analyses as a means of comprehensively envisioning feature reproducibility.
Answer 4: We fully agree with this perspective: The field of radiomics/delta-radiomics exhibits weaknesses at various levels. These weaknesses can largely be summarized by the lack of a comprehensive qualification process: 1) technical performance validation; 2) clinical qualification; and 3) qualifying utilization in clinical studies (doi.org/10.1093/jnci/djy194). The first step, which involves technical performance validation, has yet to reach a consensus.
This issue is further complicated by its intersection with the ongoing debate surrounding AI technologies, for which guidance is only beginning to emerge. One such example is the Risk-Based Credibility Assessment Framework, outlined in the FDA document “Considerations for the Use of Artificial Intelligence To Support Regulatory Decision-Making for Drug and Biological Products”.
While it is reasonable to anticipate that these aspects of the field will eventually converge into a definitive, unified methodology, significant caveats remain in the evaluations currently being conducted.
Discussion:
Comment 5: Besides, the authors have also provided additional information about possible clinical significance of the radiomics-based predictive model-that is a real quite helpful thing. Clinical considerations for implementation are mentioned by the authors but they do not cover this in detail. discuss potential benefit of artificial intelligence. cite doi: 10.1007/s00405-024-08511-5.
Answer 5: This point and citations is now covered in a new “Perspective” section at the end of the manuscript where it is said: “The upcoming pivotal study is anticipated to validate the predictive performance of our pilot study. However, the radiomics-only approach presented here is just a foundational step that could be significantly enhanced by incorporating additional imaging biomarkers, such as nutritional biomarkers (e.g., L3 – Skeletal Muscle Index) [41], as well as non-imaging biomarkers (e.g., Krebs von den Lungen-6 (KL-6)) [42]. Furthermore, advancements in AI models and the integration of complementary technologies could further elevate the system's capabilities [43].”
Comment 6: Insightful discussion of challenges in utilizing the tumor-specific models for a patient-level predictive system. More specific recommendations-for example, hierarchical or multi-level modeling approaches that might surmount some of the challenges faced by future studies.
Answer 6: (L474-477) In “Radiomics robustness” section we emphasize the need to transition from a tumor-center system to a patient-centered system in adding: “and for the final system to be practical and widely applicable, it must transition from a tumor-centered approach to a patient-centered one. This would require the implementation of hierarchical or multi-level modeling to ensure the system can address the broader complexities of patient care.”
Future Directions:
Comment 7: The authors went further to propose some relevant next steps, including the validation of the predictive model in an independent cohort and the use of radiomics/delta-radiomics in monitoring response to their treatment over time. Furthermore, the authors explored the possibility of integrating radiomics features with other clinical or molecular variables to enhance predictive performance.
Answer 7: We add a “Perspective” section at the end of the Discussion paragraph. In this section (L569-575) we list some tracks for improving the clinical benefits supported by several references, notably the one by Maniaci et al.: “The upcoming pivotal study is anticipated to validate the predictive performance of our pilot study. However, the radiomics-only approach presented here is just a foundational step that could be significantly enhanced by incorporating additional imaging biomarkers, such as nutritional biomarkers (e.g., L3 – Skeletal Muscle Index) [41], as well as non-imaging biomarkers (e.g., Krebs von den Lungen-6 (KL-6)) [42]. Furthermore, advancements in AI models and the integration of complementary technologies could further elevate the system's capabilities [43].”
Comment 8: Authors could have emphasized the need for developing and validating radiomics-derived, target-lesion-only-based response criteria beyond the currently accepted criteria for mesothelioma, which may more aptly address a few of the complexities of the disease process.
Answer 8: Thank you for this suggestion allowing to nicely improve the end of the added “perspective” section. We added: “Given the ability of radiomics to predict the therapeutic response of target lesions, we could also start considering evaluating the potential of radiomics for clinical trial monitoring, thereby improving the current response criteria used for mesothelioma.

Reviewer 3 Report
Comments and Suggestions for Authors
The manuscript is about a prediction model based on radiomics and delta-radiomics for the treatment response to TRuC-T 2 cell therapy in patients with mesothelioma in different anatomical areas. Thousands of features have been extracted and selected based on inter-correlation and the “10 samples per predictor” rule. The model has been evaluated with some performance indices (accuracy and AUC). The manuscript is well-written, but needs some improvements and corrections, as indicated in the comments below.
1. Some sentences do not contain a reference. For instance, the sentence at line 79 can refer to [https://doi.org/10.3390/math12091296] where some transforms were applied for a radiomic study. The sentence at line 80 can refer to [https://doi.org/10.3390/cancers15071968] where both radiomics and delta-radiomics were considered.
2. In Table 4 the bold font is missing.
3. In Line 307 sample->samples.
4. In Line 331 only the accuracy is reported and not the AUC value. Moreover, (0.1; 0.99) should be written as (95% CI:0.1; 0.99) in coherence to the rest.
5. Taking into account the small number of predictors (from 2 to 4) and for completeness, in Section 3.1 the name and type (intensity, shape, texture) of the features selected should be written and discussed.
6. Only one classifier (random forest) was employed in this study. The authors should highlight it in the limitation section and discuss why they decide to adopt this specific classifier and not others, like SVM or discriminant analysis.
7. In various parts of the manuscript some phrases are in different fonts with respect to the rest.
Author Response
Dear Sir,
Thank you for your review and comments, which will undoubtedly help improve certain aspects of our manuscript, notably the comment 6 for which we documented a new limitation of the study.
We have done our best to address your feedback, particularly regarding comments 5 and 6.
Best regards
Comment 1: Some sentences do not contain a reference. For instance, the sentence at line 79 can refer to [https://doi.org/10.3390/math12091296] where some transforms were applied for a radiomic study. The sentence at line 80 can refer to [https://doi.org/10.3390/cancers15071968] where both radiomics and delta-radiomics were considered.
Answer 1: Thank you for these suggestions, we inserted these two references.
Comment 2. In Table 4 the bold font is missing.
Answer 2: Typo has been fixed
Comment 3. In Line 307 sample->samples.
Answer 3: (L:387) We turned “10 sample per predictor” into “10 samples per predictor”
Comment 4: In Line 331 only the accuracy is reported and not the AUC value. Moreover, (0.1; 0.99) should be written as (95% CI:0.1; 0.99) in coherence to the rest.
Answer 4: (L: 418; 419) For sake of consistency, we change CI of accuracy into (95% CI: 0.1; 0.99) and add the AUC value of 0.61.
Comment 5. Taking into account the small number of predictors (from 2 to 4) and for completeness, in Section 3.1 the name and type (intensity, shape, texture) of the features selected should be written and discussed.
Answer 5: We have added additional details about the selected features, focusing primarily on pleural tumors, as these support the most relevant results. However, we have not included exhaustive details, as these features are complex and require extensive explanation for proper understanding. For example, the three radiomic features selected for predicting the response of pleural tumors in CT include:
"WVL.PERIODIC_SWT.LLH1_INTENSITY.BASED_RootMeanSquareIntensity"
"WVL.REFLECT_SWT.LHL1_INTENSITY.BASED_75thIntensityPercentile"
"WVL.PERIODIC_SWT.LLH1_INTENSITY.BASED_10thIntensityPercentile"
which necessitate elaborating on the Lifex software nomenclature and aspects of wavelet theory. We believe providing such in-depth explanations might detract from the overall focus of the manuscript.
Comment 6. Only one classifier (random forest) was employed in this study. The authors should highlight it in the limitation section and discuss why they decide to adopt this specific classifier and not others, like SVM or discriminant analysis.
Answer 6: (L548-552) we added: “Fourth, we were uncertain about the complexity and potential non-linear relationships between the features and target variables. Considering the small sample size, we aimed to minimize the risk of overfitting and the impact of noise in the data. Therefore, we chose Random Forest (RF) as a robust and flexible classifier. Although we considered testing other classification methods, comparing algorithm performance was beyond the scope of our study.”
Comment 7: In various parts of the manuscript some phrases are in different fonts with respect to the rest.
Answer 7: The whole manuscript is now formatted in Time New Roman, 12pt.

Round 2
Reviewer 3 Report
Comments and Suggestions for Authors
No comment.